# How do musculoskeletal disorders impact on quality of life in Tanzania? Results from a community-based survey

Eleanor Grieve [1], Manuela Deidda [1], Stefanie J Krauth,[2,3] Sanjura Mandela Biswaro,[4,5] Jo E B Halliday [6], Ping-Hsuan Hsieh [7], Clive Kelly,[8,9] Kajiru Kilonzo,[4,10] Kiula P Kiula [5,11], Rose Kolimba,[5] Elizabeth F Msoka [10], Stefan Siebert [12], Richard Walker,[9] Nateiya Mmeta Yongolo [5,10,13], Blandina Mmbaga,[4,5,10] Emma McIntosh,[1] NIHR Global Health Group

**Correspondence to**
Dr Eleanor Grieve;
Eleanor.Grieve@glasgow.ac.uk

## ABSTRACT

**Objectives** There are little available data on the prevalence, economic and quality of life impacts of musculoskeletal disorders in sub-Saharan Africa. This lack of evidence is wholly disproportionate to the significant disability burden of musculoskeletal disorders as reported in high-income countries. Our research aimed to undertake an adequately powered study to identify, measure and value the health impact of musculoskeletal conditions in the Kilimanjaro region, Tanzania.

**Design** A community-based cross-sectional survey was undertaken between January 2021 and September 2021. A two-stage cluster sampling with replacement and probability proportional to size was used to select a representative sample of the population.

**Setting** The survey was conducted in 15 villages in the Hai District, Kilimanjaro region, Tanzania.

**Participants** Economic and health-related quality of life (HRQOL) questionnaires were administered to a sample of residents (aged over 5 years old) in selected households (N=1050). There were a total of 594 respondents, of whom 153 had a confirmed musculoskeletal disorder and 441 matched controls. Almost three-quarters of those identified as having a musculoskeletal disorder were female and had an average age of 66 years.

**Primary and secondary outcome measures** Questions on healthcare resource use, expenditure and quality of life were administered to all participants, with additional more detailed economic and quality of life questions administered to those who screened positive, indicating probable arthritis.

**Results** There is a statistically significant reduction in HRQOL, on average 25% from a utility score of 0.862 (0.837, 0.886) to 0.636 (0.580, 0.692) for those identified as having a musculoskeletal disorder compared with those without. The attributes 'pain' and 'discomfort' were the major contributors to this reduction in HRQOL.

**Conclusions** This research has revealed a significant impact of musculoskeletal conditions on HRQOL in the Hai district in Tanzania. The evidence will be used to guide clinical health practices, interventions design, service provisions and health promotion and awareness activities at institutional, regional and national levels.

## STRENGTHS AND LIMITATIONS OF THIS STUDY

⇒ This is the first powered study of its kind in musculoskeletal disorders in Tanzania (and likely wider sub-Saharan Africa) using valid screening tools along with estimates of preference-based quality of life.

⇒ A two-stage sampling strategy was used to identify a representative sample of residents in the Hai district.

⇒ An economic questionnaire, tailored to the Tanzania context, captured quality of life measures and healthcare resource use and was administered to those with likely musculoskeletal disorders and selected controls.

⇒ A limitation of this study is that Tanzania has no value set for utilities. Instead, we used the weights generated in neighbouring Uganda to value health profiles and undertook a sensitivity analysis using the value sets from Ethiopia.

## BACKGROUND

Musculoskeletal (MSK) disorders are ranked as the second most common cause of disability worldwide.[1] These conditions cause not only clinical but also economic, societal and quality of life impacts on people's day-to-day lives.[2] Furthermore, MSK conditions often predispose and directly contribute to other comorbidities and non-communicable diseases (NCDs).[3] Epidemiological studies confirm the strong relationship between painful MSK conditions, lack of physical activity and resulting functional decline with frailty, loss of well-being and loss of independence and depressive symptoms.[4,5] MSK health is critical for people's mobility and dexterity, their ability to work and actively participate in all aspects of life and to maintain economic, social and functional independence across their life course.[2] MSK conditions have

suffered from a distinct lack of attention from the medical profession, politicians and the public because they are not usually life-threatening and many are considered to be a natural part of ageing, despite being the most prevalent causes of disability, absence from occupational activities and early retirement impacting both the individual and also wider society.[6]

However, little evidence exists on the prevalence, patterns and impacts of such conditions among the general population in sub-Saharan Africa (SSA).[2] Further, existing strategic plans to reduce NCDs in this setting often overlook the impact of MSK disorders.[7] When prioritising healthcare research and investment, the burden of MSK disorders has been underestimated and even ignored, mainly due to their chronic nature and perceived low fatality rate.[8] While the importance and severity of the emerging NCD burden is well recognised with strategic national action plans in place for the prevention and control of NCDs, the focus is dominated by conditions such as cardiovascular diseases, cancer and diabetes.[9] Little attention has been paid to MSK conditions. This is out of proportion to the impact that MSK conditions have on disability, with MSK being one of the major causes of non-traumatic disability within the global burden of disease.[10] Arthritis (joint diseases) is among the most common MSK disorders and is a leading cause of physical disability and severe long-term pain.[11] Measuring the full impact of these disorders is urgently needed to inform policy.

Tanzania has made progressive commitments towards Universal Health Coverage with the scale-up of health insurance to be rolled out in 2023 and has launched a National NCDs strategic plan. However, plans often prioritise other NCD conditions such as diabetes, hypertension and sickle cell disease while leaving MSK conditions behind. The 'NIHR Global Health Research Group on estimating the prevalence, quality of life, economic and societal impact of arthritis in Tanzania: a mixed-methods study at the University of Glasgow was established with the aim of characterising and estimating the various burdens of arthritis in Tanzania. This wide-ranging study aimed to establish the prevalence of arthritis and joint problems and to better understand their impact—clinically, economically and socially—in Tanzania. Here, we present the findings related to the impact on health-related quality of life (HRQOL) for those with MSK conditions and arthritis in Tanzania.

## METHODS
A community-based cross-sectional survey conducted in 15 villages in the Hai District, Kilimanjaro was undertaken between January 2021 and September 2021 working in partnership with the Kilimanjaro Clinical Research Institute. Clinical screening tools, including the Gait Arms Legs Spine (GALS)[12] and Regional Examination of the Musculoskeletal System (REMS),[13] were used to screen people for MSK conditions through a tiered approach. A two-stage cluster sampling with replacement and probability proportional to size was used to select a representative sample of the population.[14] We administered a battery of instruments including economic (ie, usage of healthcare resources; productivity impact, etc) and HRQOL questions to a sample of residents (aged over 5 years old) in selected households (N=1050 households). Questions on healthcare resource use, expenditure and quality of life were administered to all participants, with additional more detailed (long version) economic and quality of life questions administered to those who screened positive on REMS (only GALS positive individuals were assessed with REMS), indicating probable arthritis.

Each person was assessed at their home by two interviewers using the standardised study protocol.[15] The interview commenced with a series of questions prior to the performance of first GALS, then REMS (if GALS was positive). If REMS was positive, then further investigations including blood tests and X-rays were subsequently arranged as indicated. Regarding comorbidities, we asked whether the person had any previous illnesses or required previous hospitalisations. We asked about any medication they took. We also asked specifically about other common conditions including NCDs such as high blood pressure, heart disease, breathlessness, lung disease, urinary issues, kidney disease, as well as about infections such as malaria, tuberculosis and HIV.

In order to establish a reference population for these measures, this group was matched on age (±1 year for minors and ±3 years for adults) and gender, aiming for two controls per REMS positive case from the population of GALS negative participants. In cases where no age match could be identified within this range, for example, for very old participants, larger age differences were accepted rather than not recruiting a control. Hence, responses from participants screening positive for MSK conditions were able to be directly compared with a matched control group in a bid to establish the relative magnitude of quality of life impact of the MSK disorders. Full details are provided in the study protocol.[15]

### Quality of life tools
The Swahili version of the preference-based tool EuroQol 5-Dimensional 5 Level (EQ-5D-5L) was used for those 18 years and over.[16] The EQ-5D is a generic questionnaire relating to mobility, self-care, usual activity, pain and discomfort, and anxiety and depression. The EQ-5D health state is converted into a utility score (typically between 0 and 1 where 0 represents death and 1 represents full health) using a country-specific scoring algorithm or value set. The value attached to an EQ-5D profile according to a set of weights reflects on average, people's preferences about how good or bad the state is. As Tanzania has no set of weights for the EQ-5D dimensions, the Ugandan tariff was used as the base case and the Ethiopian tariff as a sensitivity analysis.[17 18] The Child Health Utility instrument was used for children and adolescents under 18 years.[19]

## Model

A generalised linear model with appropriate family/link was fitted to estimate differences in utility scores (dependent variable) between those presenting as REMS positive, that is, with probable MSK disorder and controls. The modified Park test was conducted to choose the best distribution while a suite of tests was run to guide the best choice of link. See online supplemental file 1 for full details. The primary explanatory variable categorised individuals into MSK condition (REMS+) or control groups. The regressions were also adjusted by demographics (age and gender), socioeconomic (education, occupation and marital status) and clinical/lifestyle (family members who had experienced joint pain, smoking and drinking habits, diagnosis of diabetes) variables. For education, participants were asked 'What is the highest level of school that you completed: primary/middle/higher?' Lifestyle issues (drinking, smoking habits) were asked if current, former or never. Explanatory variables in the final model included age, occupation, marital status, gender, education, comorbidities and behavioural risk factors. Although a matched control group was established, we still controlled for sex and age in our case–control model to ensure against imperfect matching.[20] SEs were adjusted for clustering at household and village levels. Multiple imputation procedures using chained equations and all covariates were used to impute missing data separately for each group (MSK/controls), creating 40 imputed datasets.[21 22] Both complete case and multiple imputation analyses were undertaken. All analyses were performed using Stata V.17.

## Patient and public involvement

The community was involved throughout this research, from design to dissemination. Please see the report which details the dissemination outputs, stakeholder engagement activities and their underlying principles conducted by this study.[23]

## RESULTS

There were a total of 594 respondents, of whom 153 had a confirmed MSK disorder and 441 matched controls. Table 1 provides a descriptive analysis comparing REMS positive individuals and matched controls, in terms of demographic, clinical and socioeconomic characteristics. Almost three-quarters of those identified as REMS positive were female and had an average age of 66 years. They were twice as likely not to be working than those without an MSK disorder (12% compared with 6%). Comorbidity was higher among REMS+patients; they were more likely to have other conditions, in particular, hypertension (24% compared with 14%). Quality of life utility scores by age and count of comorbidities showed a clear distinction between those with an MSK disorder, compared with those without an MSK disorder for all age groups.

In terms of missing data, although only one person did not answer the EQ-5D questionnaire, an 18% rate of missingness in the covariates (especially history of family pain) required the use of a multiple imputation approach. Two children were positive with GALS, one refused to be examined further with REMS and the other was REMS positive.

Table 2 provides the results for both the complete case and multiple imputed datasets using the Ugandan and Ethiopian tariffs. Using the chosen family of Gaussian non-linear least squares with identity link and taking the Ugandan results as the base case, there is a statistically significant reduction in EQ-5D HRQOL, on average 25% from a utility score of 0.86 to 0.64, for those identified as having an MSK disorder compared with those without. Age, history of family pain and no longer working are associated with a statistically significant reduction in quality of life in both the complete case and multiple imputed analyses. Gender was not a significant factor. While women are more likely to have an MSK disorder, there was no significant difference between men and women in terms of reduced HRQOL. Comorbidities do not have a statistically significant impact on HRQOL. This is likely due to the relatively low number of comorbidities in our data.

Using the Ethiopian value set for the purposes of sensitivity analyses, there is again a statistically significant reduction in quality of life, on average of around 10% from a utility score of 0.95 to 0.85, for those identified as REMS positive compared with those who are not. Absolute utility values across both groups are higher using the Ethiopian value set as compared with those generated by the Ugandan value set. Age, history of family pain and no longer working are similarly associated with a statistically significant reduction in quality of life in both the complete case and multiple imputed analyses.

All five dimensions of the EQ-5D—mobility, self-care, usual activity, pain and discomfort, and anxiety and depression—were scored lower by those with a positive diagnosis. Notably, 'pain and discomfort' was a major contributor to this reduction in quality of life. The distribution of each HRQOL dimension (for the Ugandan tariff only) is shown in figure 1. The controls (in red) show a similar distribution across each domain, skewed towards the scores indicating 'none' or 'mild' problems. In contrast, the pattern for those with MSK indicates greater numbers scoring moderate and severe problems in each domain.

## DISCUSSION

Our findings reveal a major reduction in HRQOL for people living with MSK disorders as identified through REMS screening in comparison to those without MSK disorders in the Hai district of Tanzania. A good quality of life refers to a person's ability to look after themselves, get around their community, participate in their usual activities and avoid pain and distress. Many MSK disorders such as rheumatoid arthritis disproportionately affect women.[24] Here, we find those presenting with MSK conditions to be predominantly female and to have a

**Table 1** Descriptive summary of characteristics for all participants, and by musculoskeletal/arthritis positive and matched controls

| Categories | All (n=594) | REMS+ (n=153) | Controls (n=441) |
|---|---|---|---|
| **Background characteristics** | | | |
| Age (mean) | 62.8 | 66.4 | 61.5 |
| Gender (female) | 70% | 74% | 68% |
| Occupation (farmer) | 78% | 78% | 78% |
| Unemployed or retired | 7.24% | 11.8% | 5.67% |
| Religion | | | |
| Christian | 82% | 79% | 84% |
| Muslim | 18% | 21% | 16% |
| Education | | | |
| Basic | 87% | 91% | 86% |
| Middle | 9% | 8% | 10% |
| Higher | 4% | 1% | 4% |
| Married | 58% | 49% | 61% |
| Smoker | | | |
| Current | 7% | 5% | 8% |
| Former | 9% | 10% | 9% |
| Never | 84% | 85% | 83% |
| Alcohol drinker | | | |
| Current | 31% | 32% | 31% |
| Former | 23% | 21% | 24% |
| Never | 46% | 47% | 45% |
| History of family pain | 9% | 25% | 3% |
| **Other conditions** | | | |
| +1 any other condition (n) | 18% (96) | 26% (39) | 15% (57) |
| +2 any other conditions (n) | 3% (18) | 3% (5) | 3% (13) |
| Hypertension (n) | 17% (89) | 24% (37) | 14% (52) |
| Diabetes (n) | 5% (29) | 5% (8) | 6% (21) |
| Tuberculosis (n) | 2% (8) | 1% (2) | 2% (6) |
| **Quality of life (utility weight (0–1)[42]** | | | |
| All | 0.80 (594) | 0.57 (153) | 0.88 (441) |
| By age: | | | |
| <30 years (n) | 0.94 (15) | 0.61 (2) | 0.99 (13) |
| 30–40 years (n) | 0.91 (14) | 1.0 (2) | 0.90 (12) |
| 40–50 years (n) | 0.93 (56) | 0.84 (15) | 0.96 (41) |
| 50–60 years (n) | 0.87 (136) | 0.73 (29) | 0.91 (107) |
| 60–70 years (n) | 0.83 (191) | 0.64 (43) | 0.89 (148) |
| >70 years (n) | 0.67 (182) | 0.37 (62) | 0.82 (120) |
| By comorbidity: | | | |
| +1 condition (n) | 0.75 (96) | 0.62 (39) | 0.85 (57) |
| +2 conditions (n) | 0.68 (18) | 0.32 (5) | 0.82 (13) |

REMS, Regional Examination of the Musculoskeletal System.

**Table 2** EQ-5D scores and associated decrements for those identified positive as having a musculoskeletal disorder compared with matched controls—complete case and multiple imputation using Ugandan (base case) and Ethiopian (sensitivity analysis) tariffs

| Categories | Ugandan tariff | | | | Ethiopian tariff | | | |
| | Quality of life decrements (complete case) N=488 | | Quality of life decrements (multiple imputation) N=594 | | Quality of life decrements (complete case) N=488 | | Quality of life decrements (multiple imputation) N=594 | |
| | QoL (95% CI) | P value | QoL (95% CI) | P value | QoL (95% CI) | P value | QoL (95% CI) | P value |
|---|---|---|---|---|---|---|---|---|
| MSK (REMS+) | -0.227 (-0.297 to -0.158) | 0.000 | -0.225 (-0.288 to -0.162) | 0.000 | -0.095 (-0.124 to -0.065) | 0.000 | -0.094 (-0.120 to -0.068) | 0.000 |
| Age | -0.005 (-0.007 to -0.003) | 0.000 | -0.005 (-0.007 to -0.003) | 0.000 | -0.002 (-0.003 to -0.001) | 0.000 | -0.002 (-0.003 to -0.001) | 0.000 |
| Gender (female) | -0.056 (-0.114 to 0.002) | 0.057 | -0.063 (-0.116 to -0.009) | 0.021 | -0.019 (-0.043 to 0.004) | 0.112 | -0.022 (-0.045 to -0.000) | 0.050 |
| Education (basic) | | | | | | | | |
| Middle | 0.024 (-0.054 to 0.101) | 0.550 | 0.020 (-0.050 to 0.090) | 0.580 | 0.010 (-0.020 to 0.040) | 0.511 | 0.009 (-0.018 to 0.036) | 0.507 |
| Higher | 0.017 (-0.088 to 0.123) | 0.744 | 0.012 (-0.083 to 0.107) | 0.807 | -0.000 (-0.049 to 0.049) | 0.993 | -0.002 (-0.047 to 0.042) | 0.919 |
| Labour (manual) | -0.041 (-0.097 to 0.014) | 0.140 | -0.037 (-0.086 to 0.012) | 0.135 | -0.015 (-0.037 to 0.008) | 0.202 | -0.013 (-0.033 to 0.007) | 0.193 |
| Not working | -0.282 (-0.449 to -0.114) | 0.001 | -0.281 (-0.432 to -0.129) | 0.000 | -0.108 (-0.177 to -0.039) | 0.002 | -0.107 (-0.171 to -0.044) | 0.001 |
| Married | 0.041 (-0.014 to 0.096) | 0.121 | 0.045 (-0.004 to 0.094) | 0.074 | 0.016 (-0.009 to 0.042) | 0.206 | 0.018 (-0.005 to 0.040) | 0.123 |
| Family pain | -0.197 (-0.335 to -0.059) | 0.005 | -0.208 (-0.339 to -0.077) | 0.002 | -0.076 (-0.137 to -0.016) | 0.014 | -0.079 (-0.136 to -0.023) | 0.006 |
| Smoker (never) | | | | | | | | |
| Former | -0.021 (-0.103 to 0.062) | 0.626 | -0.043 (-0.125 to 0.039) | 0.301 | -0.007 (-0.046 to 0.031) | 0.701 | -0.018 (-0.056 to 0.021) | 0.365 |
| Current | -0.088 (-0.212 to 0.035) | 0.171 | -0.086 (-0.200 to 0.029) | 0.142 | -0.037 (-0.085 to 0.011) | 0.127 | -0.036 (-0.081 to 0.008) | 0.117 |
| Alcohol drinker (never) | | | | | | | | |
| Former | -0.066 (-0.141 to 0.008) | 0.082 | -0.059 (-0.124 to 0.006) | 0.075 | -0.027 (-0.059 to 0.005) | 0.094 | -0.025 (-0.052 to 0.003) | 0.078 |
| Current | -0.009 (-0.059 to 0.041) | 0.717 | -0.023 (-0.069 to 0.023) | 0.328 | 0.000 (-0.021 to 0.021) | 0.980 | -0.006 (-0.026 to 0.014) | 0.530 |
| Diabetes | 0.034 (-0.074 to 0.143) | 0.537 | 0.032 (-0.165 to 0.128) | 0.521 | 0.021 (-0.023 to 0.056) | 0.357 | 0.017 (-0.023 to 0.056) | 0.411 |
| Hypertension | 0.028 (-0.054 to 0.110) | 0.503 | 0.021 (-0.054 to 0.096) | 0.590 | 0.014 (-0.022 to 0.050) | 0.443 | 0.010 (-0.022 to 0.043) | 0.540 |
| Constant | 1.282 | | 1.281 | | 1.109 | | 1.110 | |
| QoL utility score | REMS+ve 0.623 (0.562 to 0.684) | | REMS+ve 0.636 (0.580 to 0.692) | | REMS+ve 0.845 (0.820 to 0.870) | | REMS+ve 0.850 (0.827 to 0.874) | |
| | Controls 0.850 (0.823 to 0.878) | | Controls 0.862 (0.837 to 0.886) | | Controls 0.940 (0.928 to 0.952) | | Controls 0.945 (0.934 to 0.955) | |

EQ-5D, EuroQol 5-Dimensional; MSK, musculoskeletal; QoL, quality of life; REMS, Regional Examination of the Musculoskeletal System.

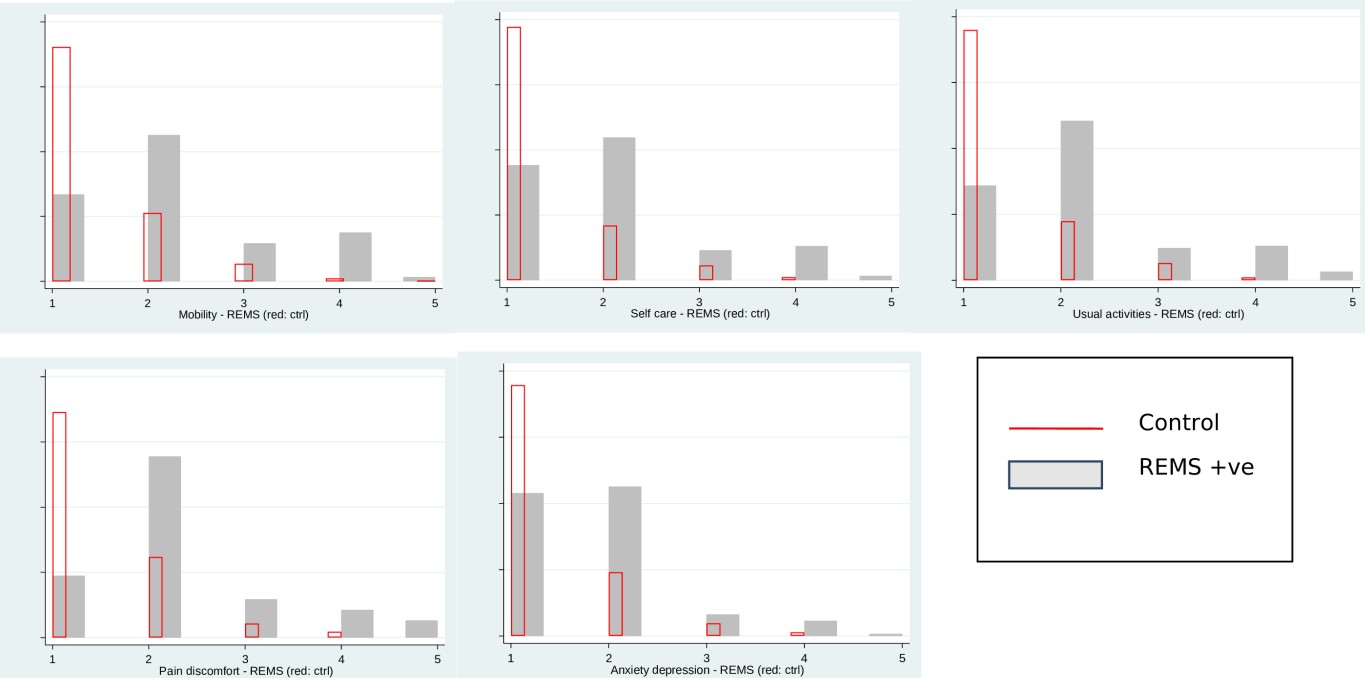

**Figure 1** Score distributions for those identified positive as having a musculoskeletal disorder compared with matched controls for each HRQOL dimension (Ugandan tariff). HRQOL, health-related quality of life; REMS, Regional Examination of the Musculoskeletal System.

significant reduction in HRQOL scores, which declines further with age. It is well documented that MSK disorders incur considerable costs to society through lost productivity,[25] and in this study 'not working' (anymore) is associated with a statistically significantly negative impact on HRQOL too. Comorbidity was also higher among REMS+participants and was largely due to their greater prevalence of hypertension. Inflammatory joint diseases and gout are linked to a greater prevalence of cardiovascular disease, and hypertension is often the first and most readily observed feature of this tendency.[26]

REMS is designed to define the cause of pain or disability in a specific region of the body. This instrument can identify and differentiate between inflammatory joint disorders, such as rheumatoid arthritis, and more common mechanical conditions (such as back pain or osteoarthritis). Furthermore, it can distinguish different types of inflammatory disorders such as psoriatic arthritis from rheumatoid arthritis, although additional radiological or blood tests may be needed to confirm these findings. The country's healthcare policy should focus on those disorders which cause the greatest loss of independence and for which intervention is affordable and effective. Figure 1 shows how each of the different dimensions of HRQOL—pain, mobility, usual activities, self-care—is negatively impacted in those identified positively as having an MSK disorder compared with matched controls. Given that low back pain is a common disability and is eminently treatable, resources to support and treat people with this ailment should be made more widely available.

Although our study was designed with the intention of capturing the impact of both MSK disorders and any associated comorbid conditions, there was the potential for certain comorbidities to be overlooked. Most commonly, these would relate to conditions for which few symptoms would be evident, such as renal disease, which is a not uncommon association of gout and autoimmune rheumatic diseases. That they are also more likely to have other conditions has led researchers to study associated patterns of multimorbidity in those with arthritis.[27] The reduction of 0.22 utilities in HRQOL found in our study between those screened positive for MSK disorders and those who did not is of similar magnitude to the reduction in quality of life for people with three or more long-standing conditions living in areas of high socioeconomic deprivation in Scotland compared with the general population.[28] In that study, Lawson *et al* observed a similar utility decrement of 0.21. The fact that the drop in utility found due to MSK disorders is comparable to that of people living with multiple long-term conditions highlights the importance of MSK health for the quality of life of patients.

Evidence from other countries shows a wide range of utility reductions in those with rheumatoid arthritis. For example, Hoshi *et al*[29] find studies range from utility scores of 0.49 to 0.75, likely due to differences in patient populations and study designs, with more recent studies having higher HRQOL values, likely reflecting improved management of patients over the last decade. A systematic review of HRQOL scores in rheumatoid arthritis from Asia found those with moderate to high disease activity scores had EQ-5D utilities of 0.53 and 0.47 respectively, which were positively associated with age.[30] However, utility values significantly differed between the countries

in which studies were conducted, including India averaging 0.40, Taiwan 0.67 and Thailand 0.81. They also found EQ-5D utility values significantly differed between study designs, with EQ-5D values for case-control of 0.65 (0.49–0.80), cohort studies 0.60 (0.54–0.66) and cross-sectional studies 0.72 (0.68–0.75).[30] The use of other HRQOL tools to generate utilities may also produce variation. Uhlig *et al.* found a utility reduction using the Short Form Survey 36 of 0.20 in patients aged 50–60 years with MSK conditions compared with the general population in Norway.[31]

Different methodologies to produce value sets may also account in part for variation in results.[32] A limitation of this study is that Tanzania has no value set for utilities. Instead, we used the weights generated in neighbouring Uganda to value health profiles. We have, for this reason, undertaken a sensitivity analysis using the value sets from Ethiopia, another East African country. Although each country employed a combination of slightly different methods, both value sets included the use of time trade-off to generate weights to health profiles where respondents choose between living in a certain impaired health state for a certain length of time, or in full health for a shorter time period.[33] In the Ugandan value set, 'pain and discomfort' was the most important dimension with a decrement of 0.798 for the most extreme level. In Ethiopia, 'pain and discomfort' was the second highest contributor to HRQOL reductions, but with a relatively smaller decrement of 0.406. MSK disorders include a range of different conditions and affect people differently, with pain being the most commonly reported symptom, although highly subjective and variable. In studies exploring HRQOL across multiple chronic diseases, pain has been found to have the largest negative impact on HRQOL[34] with MSK disorders associated with the largest losses of HRQOL.[35] Carreño *et al*[36] found 'pain or discomfort' to be the highest scoring dimension, with a maximum score in over 12.3% of cases. Our study shows a similar trend with the highest number of extreme scores found in the 'pain or discomfort' and 'mobility' domains equally. Given 'pain or discomfort' is also the most important dimension in the value set used,[18] this dimension is driving utility values on both fronts. With such a baseline figure, any future interventions aimed at alleviating symptoms including pain, or indeed any preventive interventions, will likely improve this HRQOL profile.

The dimension 'anxiety and depression' is another area of notable contrast between the two countries' value sets. In Uganda, 'anxiety and depression' was found to be the least valued on health. In contrast, in Ethiopia, 'anxiety and depression' was rated the most valued. The authors of the Ugandan study note that in Ugandan culture, it is generally considered a weakness for someone to show signs of mental illness or even the need for mental healthcare.[18] Indeed, local clinicians in Tanzania have found while collecting HRQOL data in patients that no-one will score high (severely, extremely impacted) for 'anxiety and depression' given the stigma associated with either recognising or talking about such issues publicly.[37] The fact that mental health is traditionally 'taboo' to talk about in some cultures and religions is likely to affect the scoring for the 'anxiety and depression' dimension in countries such as Tanzania. While the overall population norms in our study approximate the utility ranges for corresponding age groups from a range of other countries,[38] it is notable that our controls have a consistently higher utility score across age bands than might otherwise have been expected, especially in the older age groups. In our study, 'anxiety and depression' had the least important scores, and it is likely that utility values are therefore overestimated in this population.[39] That cultural norms may influence a person's willingness to disclose their difficulties is a common limitation of many international studies, and ours is no exception. Hence, the true burden of disease may be greater than we have reported.

Good practice would dictate that when analysing international studies, country-specific value sets should be used to generate utilities valid for that country. In various country comparisons, systematic differences have been found as well as discrepancies when analysing changes from one health state to another health state, not just in terms of magnitude but even in the direction of the change.[32] Within these East Africa value sets, generated using similar methods, it is more likely that cultural and contextual reasons are driving a variance in results. In Uganda, the mean for the study sample generating the value set was 0.86 compared with 0.94 in Ethiopia. The authors of the Ethiopian study note evidence of clustering of values at 1 with non-trading of time for milder health states.[17] This is evident in the distribution of utilities (see online supplemental file 2). Overall, compared with the Ethiopian value set, the extreme health states had lower values in Uganda. To overcome these limitations and as part of our wider National Institute for Health and Care Research (NIHR) study, there is ongoing research to develop an African-specific measure of health and well-being. This measure will be based on the capability approach,[40] a multidimensional approach to well-being, to understand what aspects of life are important to people in Tanzania using values from people in Tanzania. Arthritis has the potential to affect many aspects of the capabilities that are important to people. When completed, this measure can then be used to provide information to decision-makers about which health and care interventions will most benefit people's lives in Tanzania.

Despite these methodological limitations and varying magnitude of results, it is evident that the impact of MSK conditions on HRQOL is considerable. Decision-makers need information on both costs associated with a condition and the health improvements that can be achieved by intervening for those affected by MSK disorders when planning for cost-effective services. Working across different disciplines, and in both clinical and community settings, our wider NIHR research project has estimated the prevalence of MSK disorders, its economic burden

and impact on HRQOL.[41] In doing so, we provide an initial evidence base to justify the need to plan effective and cost-effective interventions for the prevention and management of arthritis and MSK conditions in Tanzania and wider SSA.

## CONCLUSIONS

This study quantifies the significant HRQOL burden caused by MSK disorders in Kilimanjaro, and likely wider Tanzania. This is the first powered study of its kind in MSK disorders in Tanzania (and likely wider SSA) using valid screening tools along with estimates of preference-based quality of life. The results will be used to guide clinical healthcare delivery, interventions design, service provisions and health promotion and awareness activities both at institutional level, Kilimanjaro Region and national level. The findings from this study will inform those leading the NCD strategies in Tanzania of the magnitude of the impact of arthritis and related MSK conditions to plan properly for the management of these disorders, develop bespoke interventions to prevent and reduce the impact of these conditions and to improve quality of life for those who live with and suffer from them.

**Author affiliations**
[1]School of Health and Wellbeing, Health Economics and Health Technology Assessment, University of Glasgow College of Medical Veterinary and Life Sciences, Glasgow, UK
[2]School of Health and Wellbeing, College of Medical, Veterinary and Life Sciences, University of Glasgow, Glasgow, UK
[3]School of Nursing, Midwifery, Allied and Public Health, Canterbury Christ Church University, Canterbury, UK
[4]Kilimanjaro Christian Medical Centre, Moshi, Tanzania, United Republic of
[5]Kilimanjaro Clinical Research Institute, Moshi Urban, Kilimanjaro Region, Tanzania, United Republic of
[6]School of Biodiversity, One Health and Veterinary Medicine, University of Glasgow College of Medical Veterinary and Life Sciences, Glasgow, UK
[7]National Defense Medical University, Taipei City, Taiwan
[8]James Cook University Hospital, Middlesbrough, UK
[9]Population Health Sciences Institute, Newcastle University, Newcastle upon Tyne, UK
[10]KCMC University, Moshi Urban, Kilimanjaro Region, Tanzania, United Republic of
[11]Rural Development and Regional Planning, Institute of Rural Development Planning, Dodoma, Tanzania, United Republic of
[12]School of Infection and Immunity, University of Glasgow College of Medical Veterinary and Life Sciences, Glasgow, UK
[13]Department of Clinical Sciences, Liverpool School of Tropical Medicine, Liverpool, UK

**Acknowledgements** We are grateful for all the support received for this research, including that from our study participants.

**Collaborators** The NIHR Research Group comprises (in alphabetical order): Sanjura Biswaro, Christopher Bunn, Edith Chikumbu, Jo Coast, Mia Crampin, Manuela Deidda, Nateiya M Yongolo, Eleanor Grieve, Jo Halliday, Victor Katiti, Clive Kelly, Kajiru Kilonzo, Emma Laurie, Emma McIntosh, Blandina Theophil Mmbaga, Gloria Moshi, Elizabeth Msoka, Febronia Shirima, Stefan Siebert, Jennika Virhia, Richard Walker and Sally Wyke.

**Contributors** Conception and design of the study: BM and EM. Data acquisition: SJK, NMY, SMB, JEBH, RW and BM. Data analysis: EG, MD, SJK and P-HH. Data interpretation: EG, MD, SJK, JEBH, CK, SS, RW, NMY, RK and EM. Drafting of the first version of the manuscript: EG. Critical review of the manuscript: all authors read and approved the final manuscript. EG is the guarantor for this article.

**Funding** This study/project was funded by the UK's NIHR Global Health Research Groups Programme (Award no: 17/63/35, https://fundingawards.nihr.ac.uk/award/17/63/35).

**Disclaimer** The views expressed are those of the author(s) and not necessarily those of the NIHR or the Department of Health and Social Care.

**Competing interests** None declared.

**Patient and public involvement** Patients and/or the public were involved in the design, or conduct, or reporting, or dissemination plans of this research. Refer to the Methods section for further details.

**Patient consent for publication** Not applicable.

**Ethics approval** The study was conducted in accordance with national and international ethical guidelines and principles and received clearance from the College of Medical, Veterinary & Life Sciences Ethics Committee at the University of Glasgow (Ref. number: 200180100), the Kilimanjaro Christian Medical University College Local Ethical Review Committee (KCMC/P.I/Vol.XI), and the National Institute for Medical Research in Tanzania (NIMR/HQ.R.8a/Vol.IX/3038). Written informed consent was obtained from all adult participants and guardians of participants aged <18 years after detailed information about the study was provided and time allowed for any questions. In addition to guardian consent, assent was obtained from participants aged 12–17 years. Consent documents were all provided in Swahili. In case of illiteracy or other language requirements, a witness and/or translator could be chosen by the participants to provide verbal translation during consenting and survey delivery. All data collection instruments were translated to Swahili and adapted to the local context.[43 44]

**Provenance and peer review** Not commissioned; externally peer reviewed.

**Data availability statement** Data are available on reasonable request. MD and SK have accessed and checked the underlying data. The data that support the findings of this study are available on reasonable request.

**ORCID iDs**
Eleanor Grieve https://orcid.org/0000-0002-4115-2882
Manuela Deidda https://orcid.org/0000-0002-0921-6970
Jo E B Halliday https://orcid.org/0000-0002-1329-9035
Ping-Hsuan Hsieh https://orcid.org/0000-0002-0430-260X
Kiula P Kiula https://orcid.org/0000-0002-6860-5474
Elizabeth F Msoka https://orcid.org/0000-0002-2352-3520
Stefan Siebert https://orcid.org/0000-0002-1802-7311
Nateiya Mmeta Yongolo https://orcid.org/0000-0002-9659-7926

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
