## [Reviewer comments · BMJ Open]

ARTICLE DETAILS

Title (Provisional)

How do Musculoskeletal Disorders impact on Quality of Life in Tanzania? Results from a Community-Based Survey

Authors

Grieve, Eleanor; Deidda, Manuela; Krauth, Stefanie; Biswaro, Sanjura Mandela; Halliday, J E.B.; Hsieh, Ping-Hsuan; Kelly, Clive; Kilonzo, Kajiru; Kiula, Kiula P.; Kolimba, Rose; Msoka, Elizabeth F.; Siebert, Stefan; Walker, Richard William; Yongolo, Nateiya Mmeta; Mmbaga, Blandina; McIntosh, Emma; NIHR Global Health Group, On behalf of

VERSION 1 - REVIEW

Reviewer	1
Name	Lizana, Pablo
Affiliation	Pontificia Universidad Catolica de Valparaiso, Laboratory of Epidemiology and Morphological Sciences, Instituto de Biología
Date	13-Oct-2024
COI	None

In the introduction section, the writing goes from the most general to the most particular in the population of Sub-Saharan Africa. However, the authors describe several sentences (perfect sentences) but do not have references to support these comments. For example:

1. These conditions cause not only clinical but economic, societal, and quality of life impacts on people's

day-to-day lives.

2. Furthermore, musculoskeletal conditions often predispose and directly contribute to other comorbidities and non-communicable diseases (NCDs).

3. Further, existing strategic plans to reduce NCDs in

this setting often overlook the impact of musculoskeletal disorders.

4. When prioritising healthcare

research and investment, the burden of musculoskeletal disorders has been underestimated and even

ignored, mainly due to their chronic nature and perceived low fatality rate.

5. Whilst the importance and severity of the emerging NCD burden is well recognised with strategic national action plans in place

for the prevention and control of NCDs, the focus is dominated by conditions such as cardiovascular

diseases, cancer and diabetes.

6. Arthritis (joint diseases) are among the commonest musculoskeletal disorders and are a leading cause

of physical disability and severe long-term pain

Another relevant aspect of the introduction is the mention of arthritis. However, this concept needs to be observed in the title.

Methods

The authors describe the tools used to assess quality of life and the restrictions they present. In this sense, the authors should indicate the internal consistency values presented in the questionnaires (e.g., Cronbach's alpha).

The details of the ethical considerations are interesting, considering the language barriers and illiteracy of the sample.

Results

Table 1 is attractive because it presents much information about the participants. However, much of this information needs to be justified regarding methods. For example, why only the occupation of a farmer is presented (remember that this journal is international and not everyone knows the sociodemographic characteristics of the country), only gender was asked in a dichotomous way, why is the type of religion asked, is religion relevant in musculoskeletal disorders, education should be described in years of schooling, and why is religion relevant in musculoskeletal disorders? Is religion relevant in musculoskeletal disorders? Education should be described in years or indicate what is meant by "basic," "medium," and "higher." Why are only married people asked? They have to describe why they are asked about smoking behavior and also about drinking alcoholic beverages. That part of the table indicating other health conditions should be better explained. Table 2 even includes the variable "ever pregnant," which has not been described or justified in the methods and is not presented in Table 1.

General comments,

Several variables not described in the methods are presented in tables and are unjustified.

The authors describe this as one of the first studies on musculoskeletal disorders. In that sense, the authors do not present musculoskeletal disorders by body segment. Should the country's public policies focus on musculoskeletal disorders of the lower back or on neck/shoulder disorders?

It is interesting that the authors discuss dimensions of quality of life. However, methods and results still need to be developed. In other words, in discussions, it is noted that each dimension of quality of life is relevant. Therefore, the authors should describe and analyze them.

Reviewer	2
Name	Svensson, Mikael
Affiliation	Health Metrics, Sahlgrenska Academy at University of Gothenburg & Centre for Health Economics (CHEGU) at University of Gothenburg
Date	23-Jun-2025
COI	None

This is a short, to-the-point study that is nicely structured and easy to follow. But, I do think it is a bit too short on details and discussions, and I have a few comments that relate to additional details that I would like to see in the paper.

* Methods: For readers not having any previous experience with the REMS instrument, additional details on what type of musculoskeletal disorders that are being captured (mostly) would be relevant. Since MD can include everything from severe RA to "mild back pain", what is it that is generally captured here in this population group?

* Methods: I would suggest that additional details on the MI approach are included (were all covariates used in the process?).

* Methods: Data collection was carried out by interviews? How many interviewers were used or how was the process set out? Even though there is a reference to a study protocol, I think some additional details on the data collection should be included in the study.

* Methods: I think there also needs to be more information on comorbidities. What questions were used to screen for the prevalence of other conditions (list in supplement perhaps?).

* Results: The share of comorbidities is higher in the REMS+ group (1.73x), is this primarily driven by the difference in the prevalence of hypertension? This goes back to my previous comments (on Methods), in which I would like to see more information on what type of comorbidities were asked about.

* Results: How should we understand that comorbidities seem to have no impact on the QoL decrements? Additionally, as I understand Table 2, all the comorbidity-variables are included in the same regression - could this risk "explaining away" some of the impact if they are highly correlated?

* Discussion: Coming back to comorbidities again - I am concerned as to what extent you are capturing the additional impact of MD, and whether you are also likely capturing that persons with MD also have a range of other health problems. Thus, I would like to see more discussion about this; (i) how would you explain that other comorbidities do not seem to impact QoL, (ii) what potential comorbidities may be missed in your data, (iii) is the QoL decrement reasonable considering an MD population in large (this also goes back to my comment on "what type" of MD are you primarily capturing).

* Discussion: The authors outline (p. 10) that "good practice would dictate that when analysing international studies, country-specific value sets should be used...". I was thinking about this in relation to the authors also highlighting that due to cultural norms, many (most?) people will not be ready to admit the occurrence (or impact) of mental health issues. What does such a norm imply for the relevance of HRQoL measurements (both for health profiles and value sets) using e.g. the EQ5D? The impact on "actual" QoL is likely to exist even if cultural norms imply that someone will not admit to such issues. If these biases health profiles as well as value sets, is this going to bias the study results "upwards" or "downwards"? I think this would warrant a brief discussion in the paper.

VERSION 1 - AUTHOR RESPONSE

Reviewer 1	Response	Section
In the introduction section, the writing goes from the most general to the most particular in the population of Sub-Saharan Africa. However, the authors describe several sentences (perfect sentences) but do not have references to support these comments. For example:	Thank you for highlighting this. This is well-noted. References have now been added as shown below.	
1. These conditions cause not only clinical but economic, societal, and quality of life impacts on people's day-to-day lives.	Briggs AM, Cross MJ, Hoy DG, Sánchez-Riera L, Blyth FM, Woolf AD, et al. Musculoskeletal Health Conditions Represent a Global Threat to Healthy Aging: A Report for the 2015 World Health Organization World Report on Ageing and Health. Gerontologist . 2016;56 Suppl 2:S243-55.	References
2. Furthermore, musculoskeletal conditions often predispose and directly	Koo M, Lu MC. Rheumatic Diseases: New Progress in Clinical Research and Pathogenesis. Medicina (Kaunas) . 2023 Aug 31;59(9):1581. doi:	References

contribute to other comorbidities and non-communicable diseases (NCDs).	10.3390/medicina59091581. PMID: 37763699; PMCID: PMC10534296.	
3. Further, existing strategic plans to reduce NCDs in this setting often overlook the impact of musculoskeletal disorders.	Hoy D, Geere JA, Davatchi F, Meggitt B, Barrero LH. A time for action: Opportunities for preventing the growing burden and disability from musculoskeletal conditions in low- and middle-income countries. Best Pract Res Clin Rheumatol. 2014 Jun;28(3):377-93. doi: 10.1016/j.berh.2014.07.006. Epub 2014 Oct 22. PMID: 25481422	References
4. When prioritising healthcare research and investment, the burden of musculoskeletal disorders has been underestimated and even ignored, mainly due to their chronic nature and perceived low fatality rate.	Storheim K, Zwart J. Musculoskeletal disorders and the Global Burden of Disease study. Annals of the Rheumatic Diseases 2014; 73 :949-950.	References
5. Whilst the importance and severity of the emerging NCD burden is well recognised with strategic national action plans in place for the prevention and control of NCDs, the focus is dominated by conditions such as cardiovascular diseases, cancer and diabetes.	The National Strategic Plan for the Prevention and Control of NCDs (2016–2020) centres on four major NCD groups cardiovascular diseases, cancer, chronic respiratory diseases, and diabetes as the primary targets for interventions. The current National Strategic Plan for 2021–2026 continues to highlight a high burden of NCDs particularly cardiovascular diseases, diabetes, cancer, and respiratory conditions while indicating that resource constraints challenge broader responses. Further commentary by experts in Tanzania note that although the national strategy acknowledges a diverse range of NCDs, the main focus still rests on cardiovascular diseases, cancer, and diabetes. Ref: Ngowi, J.E et al (2023) ‘Efforts to Address the Burden of Non-Communicable Diseases Need Local Evidence and Shared Lessons from High-Burden Countries’, Annals of Global Health , 89(1), p. 78.	References
6. Arthritis (joint diseases) are among the commonest musculoskeletal disorders and are a leading cause of physical disability and severe long-term pain	Safiri S, Kolahi AA, Cross M, et al. Prevalence, Deaths, and Disability-Adjusted Life Years Due to Musculoskeletal Disorders for 195 Countries and Territories 1990-2017. Arthritis Rheumatol. 2021 Apr;73(4):702-714. doi: 10.1002/art.41571. Epub 2021 Feb 22. PMID: 33150702.	References
7 Another relevant aspect of the introduction is the mention of arthritis. However, this concept needs to be observed in the title.	Thank you for this observation. We have left the title unchanged given the study’s related papers refer to musculoskeletal disorders which include arthritis. For example, this sister paper in the BMJ: https://bmjopen.bmj.com/content/15/1/e087425 .	Unchanged

Methods: 8 The authors describe the tools used to assess quality of life and the restrictions they present. In this sense, the authors should indicate the internal consistency values presented in the questionnaires (e.g., Cronbach's alpha).	As the EQ5D, and other generic quality of life tools, measure multi-dimensional aspects of health, this approach used to assess responses to the same construct is not appropriate (ref: Konerding, U. What does Cronbach's alpha tell us about the EQ-5D? A methodological commentary..". Qual Life Res 22, 2939–2940 (2013). https://doi.org/10.1007/s11136-013-0430-9). A test-retest measure may have provided an alternative means but the EQ5D was applied only once.	Unchanged
9 The details of the ethical considerations are interesting, considering the language barriers and illiteracy of the sample.	We acknowledge this comment having listed out in detail the procedures followed for obtaining informed ethics in this context.	Unchanged
Results 10 Table 1 is attractive because it presents much information about the participants. However, much of this information needs to be justified regarding methods. For example, why only the occupation of a farmer is presented (remember that this journal is international and not everyone knows the sociodemographic characteristics of the country), only gender was asked in a dichotomous way, why is the type of religion asked, is religion relevant in musculoskeletal disorders, education should be described in years of schooling, and why is religion relevant in musculoskeletal disorders? Is religion relevant in musculoskeletal disorders? Education should be described in years or indicate what is meant by "basic," "medium," and "higher." Why are only married	Thank you for these important points. For pragmatic reasons we use farmer, with most people in both groups describing themselves as this. Other categories included employed, retired, business, home help. Gender was asked as an open ended question. Religion and 'ever pregnant' are now excluded from the model as not likely to be relevant to the wider issues we address. We have defined educational attainment and lifestyle in Methods. We have added the following text to methods: "The primary explanatory variable categorised individuals into MSK condition (REMS+) or control groups. The regressions were also adjusted by demographics (age and gender), socioeconomic (education, occupation and marital status) and clinical/lifestyle (family members who had experienced joint pain, smoking and drinking habits, diagnosis of diabetes) variables". For education, participants were asked "What is the highest level of school that you completed: primary/middle/higher?" Lifestyle issues (drinking, smoking habits) were asked if current, former or never".	Methods

people asked? They have to describe why they are asked about smoking behavior and also about drinking alcoholic beverages. That part of the table indicating other health conditions should be better explained. Table 2 even includes the variable "ever pregnant," which has not been described or justified in the methods and is not presented in Table 1.		
General comments 11 Several variables not described in the methods are presented in tables and are unjustified.	See response above.	Methods
12 The authors describe this as one of the first studies on musculoskeletal disorders. In that sense, the authors do not present musculoskeletal disorders by body segment. Should the country's public policies focus on musculoskeletal disorders of the lower back or on neck/shoulder disorders?	We have added this to discussion (alongside response to Reviewer 2, comment 2): "The country's health care policy should focus on those disorders which cause the greatest loss of independence and for which intervention is affordable and effective. Given that low back pain is a common disability and is eminently treatable, resources to support and treat people with this ailment should be made more widely available".	Discussion
13 It is interesting that the authors discuss dimensions of quality of life. However, methods and results still need to be developed. In other words, in discussions, it is noted that each dimension of quality of life is relevant. Therefore, the authors should describe and analyze them.	The distribution of each HRQOL dimension shown in Figure 1 is now elaborated upon in response to Reviewer 1, comments 12-13 and Reviewer 2, comment 2)	Discussion
Reviewer: 2	Response	Section
1 This is a short, to-the-point study that is nicely structured and easy to	Thank you. Discussion has been expanded and more details added further to both reviewers valuable comments.	

follow. But, I do think it is a bit too short on details and discussions, and I have a few comments that relate to additional details that I would like to see in the paper.		
2 Methods: For readers not having any previous experience with the REMS instrument, additional details on what type of musculoskeletal disorders that are being captured (mostly) would be relevant. Since MD can include everything from severe RA to "mild back pain", what is it that is generally captured here in this population group?	We have added this to discussion (alongside response to Reviewer 1, comments 12-13): "REMS is designed to define the cause of pain or disability in a specific region of the body. This instrument can identify and differentiate between inflammatory joint disorders (such as RA) and more common mechanical conditions (such as back pain or osteoarthritis). Furthermore, it can distinguish different types of inflammatory disorders such as psoriatic arthritis from RA, although additional radiological or blood tests may be needed to confirm these findings".	Discussion
3 Methods: I would suggest that additional details on the MI approach are included (were all covariates used in the process?).	Yes, all covariates were used in the process and this has been added to Methods: "Multiple imputation procedures using chained equations and all covariates were used to impute missing data separately for each group (MSK/controls), creating 40 imputed datasets."	Methods
4 Methods: Data collection was carried out by interviews? How many interviewers were used or how was the process set out? Even though there is a reference to a study protocol, I think some additional details on the data collection should be included in the study.	Thank you, we agree and more has been added to Methods: "Each person was assessed at their home by two interviewers using the standardised protocol previously described (reference study protocol). The interview commenced with a series of questions prior to the performance of first GALS, then REMS (if GALS was positive). If REMS was positive, then further investigations including blood tests and X rays were subsequently arranged as indicated".	Methods
5 Methods: I think there also needs to be more information on comorbidities. What questions were used to screen for the prevalence of other conditions (list in supplement perhaps?).	Thank you, we agree and more has been added to Methods: "Regarding co-morbidities, we asked whether the person had any previous illnesses or required previous hospitalisations. We asked about any medication they took. We also asked specifically about other common conditions including NCDs such as high blood pressure, heart disease, breathlessness, lung disease, urinary issues, kidney disease, as well as about infections such as malaria, tuberculosis and HIV".	Methods
6 Results: The share of comorbidities is higher in the REMS+ group	Thank you, we have added more to Results: "Comorbidity was higher among REMS+ patients". Added to Discussion: "Comorbidity being higher among	Results and Discussion

(1.73x), is this primarily driven by the difference in the prevalence of hypertension? This goes back to my previous comments (on Methods), in which I would like to see more information on what type of comorbidities were asked about.	REMS+ participants was largely due to their greater prevalence of hypertension. Inflammatory joint diseases and gout are linked to a greater prevalence of cardiovascular disease and hypertension is often the first and most readily observed feature of this tendency (ref 29)".	
7 Results: How should we understand that comorbidities seem to have no impact on the QoL decrements? Additionally, as I understand Table 2, all the comorbidity-variables are included in the same regression - could this risk "explaining away" some of the impact if they are highly correlated?	Thank you for raising this point. Upon checking, the model was adjusted for specific conditions but also inadvertently the grouped variable of having 1 or > 1 condition was also included. This 'grouped' variable has been removed (and the multicollinearity this caused in the model). Statistical tests for multicollinearity have been carried out on the final model using the variance inflation factor. With an average of 1.32 mean vif, this indicates low correlations between predictors in the model. The model has been re-run and updated results are presented. Impact upon results is negligible.	Results
8 Discussion: Coming back to comorbidities again - I am concerned as to what extent you are capturing the additional impact of MD, and whether you are also likely capturing that persons with MD also have a range of other health problems. Thus, I would like to see more discussion about this; (i) how would you explain that other comorbidities do not seem to impact QoL, (ii) what potential comorbidities may be missed in your data, (iii) is the QoL decrement reasonable considering an MD population in large (this also goes back to my comment on "what type" of MD are you primarily capturing).	Again, thank you. We have added more on comorbidities to methods, discussion and limitations to address these important points. "Although our study was designed with the intention of capturing the impact of both musculoskeletal disorders and any associated comorbid conditions, there was the potential for certain comorbidities to be overlooked. Most commonly, these would relate to conditions for which few symptoms would be evident such as renal disease which is a not uncommon association of gout and autoimmune rheumatic diseases".	Discussion/ limitations
9 Discussion: The authors outline (p. 10) that "good practice	Thank you, we have added this as a limitation: "That cultural norms may influence a person's willingness to disclose their difficulties is a common limitation of	Discussion/ limitations

would dictate that when analysing international studies, country-specific value sets should be used...". I was thinking about this in relation to the authors also highlighting that due to cultural norms, many (most?) people will not be ready to admit the occurrence (or impact) of mental health issues. What does such a norm imply for the relevance of HRQoL measurements (both for health profiles and value sets) using e.g. the EQ5D? The impact on "actual" QoL is likely to exist even if cultural norms imply that someone will not admit to such issues. If these biases health profiles as well as value sets, is this going to bias the study results "upwards" or "downwards"? I think this would warrant a brief discussion in the paper.	many international studies and ours is no exception. Hence, the true burden of disease may be greater than we have reported".	
--	--	--